# Strengthening Vaccine Regulation: Insights from COVID-19 Vaccines, Best Practices, and Lessons for Future Public Health Emergencies

**DOI:** 10.3390/vaccines13060638

**Published:** 2025-06-12

**Authors:** Razieh Ostad Ali Dehaghi, Alireza Khadem Broojerdi, Alaa Magdy, Marie Valentin, Juliati Dahlan, Obaidullah Malik, Richard H. Siggers, Edwin Nkansah, Hiiti B. Sillo

**Affiliations:** 1Regulation and Prequalification Department, World Health Organization, 1211 Geneva, Switzerland; ostadalidehaghir@who.int (R.O.A.D.); magdya@who.int (A.M.); valentinm@who.int (M.V.); silloh@who.int (H.B.S.); 2National Agency of Drug and Food Control (BPOM), Jakarta 10560, Indonesia; juliati@pom.go.id; 3Drug Regulatory Authority of Pakistan, Islamabad 45500, Pakistan; dr.obaidullah@dra.gov.pk; 4Health Canada, Ottawa, ON K1A0K9, Canada; richard.siggers@hc-sc.gc.ca; 5Food and Drugs Authority, Accra CT 2783, Ghana; edwin.nkansah@fda.gov.gh

**Keywords:** regulatory preparedness, COVID-19 vaccine, Emergency Use Listing (EUL), Emergency Use Authorization (EUA), regulatory reliance, regulatory system, vaccines approval, global health governance

## Abstract

**Background**: The COVID-19 pandemic necessitated immediate regulatory vaccine approvals to facilitate timely global access. The prevailing differences in economies and resources and the varying maturity of the regulatory systems worldwide resulted in different levels of capacity to ensure vaccine quality, safety, and efficacy. In addition to the Emergency Use Authorization or equivalent by some advanced regulatory agencies, the WHO issued Emergency Use Listings (EULs), among other tools, to streamline and expedite regulatory approvals globally. This study aimed to assess the regulatory strategies and best practices adopted during the COVID-19 vaccine approvals and gather lessons for future emergency preparedness. **Methods**: A mixed-method approach employing qualitative desk reviews and a cross-sectional study collected data from 194 national regulatory authorities (NRAs) across all WHO regions. **Results**: Three main approval processes were identified: procurement-driven, reliance-based, and independent evaluations. Wealthier countries with more mature regulatory systems were found to spend a longer time issuing approvals, primarily due to being the initial assessors of the vaccines’ quality, safety, and efficacy. Furthermore, various regulatory flexibilities and best practices centered around regulatory reliance, rolling reviews, fast-tracking reviews, and employing digital tools were identified. Notably, the WHO’s EULs were essential in facilitating the timely approval of vaccines globally, including in low- and middle-income countries. **Conclusions**: The findings suggest a significant turn in vaccine regulation theories and practice, emphasizing balancing speed with scientific validity. This necessitates the creation of thorough provisions for emergency preparedness, regulatory reliance, and administrative flexibility in regulatory practices worldwide.

## 1. Introduction

The coronavirus disease 2019 (COVID-19) pandemic posed a global public health emergency nearly a decade after the H1N1 influenza pandemic 2009–2010 [1]. With over 770 million recorded COVID-19 cases and approximately 7 million deaths by 2024, every country required access to life-saving vaccines as quickly as possible. In addition to other medical products, such as personal protective equipment (PPE) diagnostics, and vaccines were essential for controlling the pandemic [1,2].

Robust regulatory systems protect the public from inadequate-quality, unsafe, or inefficacious medical products. Inefficiencies in such systems, including unnecessary delays in regulatory approvals, can directly affect access to these life-saving medical products, including vaccines [3]. The approval of the COVID-19 vaccines served as a significant test of global regulatory readiness for public health emergencies, offering valuable lessons for enhancing future responses to similar situations.

After the experience of the H1N1 influenza pandemic in 2009 and the COVID-19 pandemic in 2020, the World Health Organization(WHO) developed guidance and recommendations for national regulatory authorities (NRAs) to enable them to expedite the process of marketing authorization without compromising the quality, safety, and efficacy of the vaccine products. In the H1N1 influenza pandemic 2009, the WHO developed regulatory preparedness guidelines for the market authorization (MA) of influenza vaccines in non-vaccine-producing countries. The guidelines highlighted the steps to ensure regulatory flexibility and emphasized reliance on multi-sectoral participation [4]. By adhering to the principles contained in these guidelines, the regulatory authorities (RAs) in non-vaccine-producing countries could undertake the regulatory oversight of influenza vaccines during public health emergencies [5,6]. In September 2020, the WHO developed guidelines on “Consideration for evaluation of COVID-19 vaccines”. Revised in 2022, this document not only provides advice to manufacturers on both the process and the criteria that the WHO will use to evaluate COVID-19 vaccines that are submitted either for prequalification (PQ) or for Emergency Use Listing (EUL), but also can be used by the NRAs in proceeding to marketing authorization of these vaccines [7].

Other measures by the WHO included the development of the Quality Management System (QMS) guideline to guide Member States in improving and implementing their QMS for optimizing regulatory processes, including defining potential risks in providing regulatory oversight [8,9]. Also, in 2016, WHO initiated the Global Benchmarking Tool (GBT) to enable the determination of RAs’ capacity and the subsequent Institutional Development Plan (IDP) to improve regulatory practices, including the level of preparedness [10,11].

Typically, very short timelines are allocated for reviewers to process MA applications for medical products required to address public health emergencies. Thus, independent reviews by individual Member States potentially delay public access to these life-saving products. By promoting reliance practices, WHO aims to minimize duplication of effort and increase access to regulatory expertise among the MS. As a result, WHO released special guidelines on Good Reliance Practices in regulating medical products [12]. WHO also advocates for reliance through the Collaborative Registration Procedure (CRP), reliance on Stringent Regulatory Authorities (SRA) and WHO-Listed Authorities (WLAs), and WHO’s Emergency Use Assessment and Listing (EUL) [13,14,15]. Through these mechanisms, the WHO envisioned the assurance of timely registration of much-needed vaccine products by reducing the time needed for their regulatory approvals [16].

In the face of the COVID-19 pandemic, WHO and collaborating partners through the COVAX initiative played a key role in facilitating the regulatory approval of vaccines in its MSs. These efforts ensured timely access without compromising quality, safety, and efficacy. They included the timely provision of training to create adequate awareness among NRAs on pre- and post-authorization data and other key aspects. Also, the WHO coordinated the review process, shared information under confidentiality agreements, and guided national deployment and vaccination plans. These collaborative efforts allowed the timely issuance of in-country authorization for use and the shipping of over 1.3 billion doses of vaccines by March 2022, providing faster access to quality-assured vaccines through informed reliance practices [17,18].

This study aimed to assess the best practices and adaptive strategies employed during the regulatory approval processes for COVID-19 vaccines within the WHO and its MSs while identifying potential areas for improvement. The presented findings will assist the development of expedited registration pathways for future vaccines, improving regulatory preparedness for public health emergencies.

## 2. Methods

### 2.1. Study Design

A mixed-method study design employing thematic/content analysis (qualitative) and an analytical cross-sectional study (quantitative) was used to explore the pathways for approving COVID-19 vaccines, their flexibility, timing, and approval status, and various best practices.

### 2.2. Study Setting and Participants

In total, 194 NRAs from WHO MSs across all WHO regions participated in the study. The study was conducted between February 2021 and March 2022, during which purposefully selected key informants/contact persons from respective NRAs provided the WHO regional offices with the needed data, documents, and online resources. Total population sampling was employed to include NRAs from all MSs. Given the pandemic status, this approach aimed to ensure the maximum possible representation of practices and experiences from all the WHO regions, including economic status and the NRA’s maturity levels.

### 2.3. Data Collection Methods

The COVAX Tracker was primarily used to capture quantitative data related to vaccine regulatory approvals, timelines, and distributions. It was newly developed during the COVID-19 pandemic to provide real-time tracking of such aspects and regulatory statuses through the COVAX facility. The COVAX Tracker collected data across several variables, mainly grouped into geographic information, manufacturer details, and regulatory outcomes. This yielded structured numeric, categorical, and text-based data from close-ended responses.

Structured self-administered questionnaires and checklists were utilized to collect the necessary data from the key informants and documents. While the questionnaires were distributed to the participants for them to provide the requested information, the checklists were filled with information gathered during the desk reviews of papers and online resources. The document review was conducted by a multidisciplinary team of seven professionals from the World Health Organization (WHO), the Italian Medicines Agency (AIFA), the Pharmaceutical Management Science Association (PMSA), the Ministry of Health, Labour and Welfare of Japan, and the International Coalition of Medicines Regulatory Authorities (ICMRA). The validity and reliability of the data collection tools were evaluated through a pilot study involving the first 20 national regulatory authorities (NRAs). Further modifications were made to enhance clarity, improve flexibility, and incorporate any overlooked information following the pilot study. All key informants and respective NRAs provided informed consent and participated voluntarily.

### 2.4. Variables and Measures

The qualitative part of the study explored the major approaches to the regulatory approval of COVID-19 vaccines and the types of regulatory flexibilities employed during the approval process. Inductive thematic analysis was used to analyze the qualitative data from the desk reviews of documents. Two independent coders analyzed the qualitative data to identify recurring themes and patterns, resolving discrepancies through discussion until a consensus was reached. Themes were derived directly from the data without preconceived notions or theories.

The independent variables in the quantitative study included the World Bank economic status and the WHO-GBT maturity level (ML), whereas the dependent variable was the time of approval (TA). The independent variables were obtained from official World Bank and WHO documents, while the TA was measured as the number of days taken for the approval of various COVID-19 vaccines. Moreover, descriptive statistics were used to establish correlations between the dependent and independent variables by determining the Spearman’s rank correlation coefficient [14,18,19].

## 3. Results

A total of 194 NRAs were studied, of which the details related to their regional distribution, World Bank economic status, and WHO-GBT maturity level are as described in Table 1 below:

### 3.1. Approaches to Regulatory Approvals

The approaches to regulatory approvals undertaken by the studied Member States were found to fall into three major categories (Figure 1).

These included a procurement-driven/import-permits-only approach, whereby the authorization was granted based on the regulatory recognition of the WHO PQ, EUL, or SRA EUA/conditional MA. Depending on the local settings, the authorizations could be given by the respective NRA, Ministry of Health, or cabinet. Such countries were typically observed to issue the authorizations within 1 to 5 days, which was followed by the importation and distribution of the vaccines through the available procurement channels.

Moreover, other states followed the country-driven regulatory approach, which relied on prior assessments and data from the WHO and SRAs to grant conditional MA or EUA. This approach was reported to take 5 to 30 days to issue authorizations, and the vaccines were thereafter imported and distributed by procurement agencies or MAH.

Further, the final category included granting conditional MA or EUAs following their own assessments by the respective countries. This required the submission of the respective dossiers for COVID-19 vaccines without previous assessments by the WHO or SRAs. Special fast-track or accelerated procedures were put in place by the NRAs to perform scientific evaluations and leverage regional assessments and work-sharing initiatives. The respective MAH thereafter undertook the importation and distribution of the vaccines. Taking the WHO-European region as an example, the distribution of the Member States into the categories described above is indicated in Table 2 below.

### 3.2. Types of Regulatory Flexibilities Applied to COVID-19 Regulatory Approvals

Countries were observed to employ a range of regulatory flexibilities to enable the timely approval of COVID-19 vaccines. Most of the implemented measures covered various aspects of registration and marketing authorization, including submission, evaluation, and final authorization (Table 3). The measures were notably geared toward ensuring the timely availability of vaccines through improved regulatory efficiency and flexibility.

### 3.3. Status and Timing of Member State Approvals

Over the 16-month duration between December 2020 and March 2022, the WHO issued an Emergency Use Listing (EUL) for eleven COVID-19 vaccines of different natures from manufacturers (Figure 2). These included two mRNA vaccines, six recombinant vaccines, and three inactivated whole virus vaccines, which companies with plants in multiple locations around the globe had manufactured. In cases where an SRA has assessed the vaccines submitted for a EUL, the WHO relied on the assessments already available to conduct a risk-based evaluation of the respective products.

As of 22 February 2022, seven vaccines with EUL by the WHO had been approved and given regulatory clearance across multiple countries and territories. Leading the trend was VAXZEVRIA^®^/COVISHIELD^®^ (AstraZeneca/Serum Institute of India), which had been approved in 142 countries/territories with a total of 1485 regulatory clearances allowing the initial authorization for use and authorization for any subsequent changes (Figure 3). The vaccine also involved the most drug substance (DS) sites (eight) and drug product sites (twelve). Other vaccines with higher approval and regulatory clearance frequencies included those manufactured by Janssen-Cilag International NV and Pfizer/BioNTech Manufacturing GmbH.

As of 1 March 2022, the seven vaccines with the WHO EUL status had obtained 3291 regulatory approvals across 183 countries/territories, with the respective authorities’ target for issuing EUAs set at 14 days (Figure 4). The VAXZEVRIA^®^/COVISHIELD^®^ vaccine (AstraZeneca/SII) was the first available for distribution via the COVAX initiative. Within 15 days of its WHO EUL, it was granted import permits/approvals in 101 out of the 145 countries/territories seeking to receive vaccines via the COVAX initiative.

Among the 101 countries/territories, it was found that 28 required just an import permit for the AZ/SII vaccine before its distribution (Figure 5). Moreover, the majority (63) of the countries/territories had issued approval for the vaccine within two to seven days after listing by the WHO. By the end of the 15 days, 21 of the 145 countries had not approved the vaccine but relied on at least one expedited pathway. Further, 23 other countries had not approved the vaccine, in addition to not relying on any expedited pathway.

### 3.4. Wealth Status and Regulatory Maturity Level as Predictors of Timely Acceptance of Vaccines

A weak positive correlation (*ρ* = 0.36) (two-tailed *p* = 0.00) was observed between the four ranks of World Bank (WB) status and the time of approval of the AZ/SII vaccine. On the other hand, a moderate positive correlation (*ρ* = 0.51) (two-tailed *p* = 0.00) was observed between the WHO maturity level ranks and the vaccine’s approval time (Table 4). Therefore, the results indicate a significantly longer approval time with increased wealth and regulatory maturity.

### 3.5. Other Reported Best Practices Around Approvals of COVID-19 Vaccines

Best practices under four major categories were reported to have been adopted in response to undertaking various regulatory functions (Table 5). These included pre-planning for emergencies by having an emergency preparedness plan that prescribes clear regulatory pathways, enabling speedy responses/regulatory actions. Other best practices were reported in practicing regulatory reliance on reference authorities and institutions, increased flexibility in discharging regulatory functions, optimal coordination of key stakeholders, and approaches to sustaining regulatory efficiency post-pandemic. Given the unpredictable nature of emergencies, the adopted best practices were essential in stepping out of standard routine operations and expediting regulatory approvals and other actions.

### 3.6. Lessons Learned

Several lessons were highlighted at the global level. These include developing a standing system for coordinating global regulatory approvals and product distribution systems during public health emergencies. This will avoid delays and inefficiencies during emergencies. Moreover, a shared understanding of best practices between donors and regulators is needed to prevent donations of nearly unfit products and the donor-driven selection of recipient countries for the donated vaccines.

Also learned was the essence of balancing the urgent need for vaccines and the need to respect and uphold the national regulatory process, representation in assessment processes, and adherence to confidentiality agreements. Regarding financial and human resource flexibility, rapid hiring can manage an increased workload and minimize operational inefficiencies and staff burnout. Furthermore, global partners must build strong and functional relationships with decision-makers at the country level before emergencies to avoid the difficulty of doing so during emergencies.

## 4. Discussion

### 4.1. Approaches to Issuing Regulatory Approval for COVID-19 Vaccines by NRAs

Three main pathways to the regulatory approval of COVID-19 vaccines were adopted based on the respective country’s level of involvement/interest in undertaking independent assessments. These included a procurement-driven approach, a country-reliance approach, and a country/region-driven independent assessment approach. Other studies have reported similar pathways for COVID-19 vaccine approvals by different countries. Employing reliance mechanisms, rolling reviews, collaboration among the NRAs and the WHO, EULs, and EUAs were reported to have accelerated the approval processes and the subsequent availability of vaccines [20,21]. For example, 45% of the COVID-19 vaccines in Latin America were reported to have been approved through the reliance pathway compared to 21% of the vaccines being approved through independent assessments [20]. Strong international cooperation (among countries, regions, and other global organizations) and streamlined regulatory communication and decision-making through digital tools primarily facilitated these actions [21,22].

On the other hand, political factors, vaccine nationalism, geopolitical preferences, the lack of transparency, institutional inefficiency, bilateral procurement outside the COVAX initiative, and working in silos have been linked to slower regulatory approval processes [20,21,22]. Although many NRAs did not receive direct applications for vaccine registration or approval, the WHO served as a hub for providing access to relevant quality, safety, and efficacy data, along with appropriate assessment and evaluation reports. This crucial role enabled informed reliance and highlighted that manufacturers cannot be expected to submit applications to multiple countries at once during emergencies, emphasizing the need for the WHO to take on a coordinating role. These findings have, therefore, highlighted the critical role of international regulatory collaboration and various adaptability mechanisms to deliver efficient and highly reliable regulatory processes during emergencies. However, it is essential to acknowledge the importance of rigorous scientific evaluation in providing regulatory approval, a process that has faced criticism in certain contexts during the COVID-19 pandemic and deserves careful consideration in future vaccine approval assessments. It is important to note that, in some instances, the need for rapid authorization has restricted the ability to evaluate the long-term effects of new vaccine platforms, highlighting a crucial area for consideration in future emergency use guidelines and post-market surveillance plans.

### 4.2. Adopted Regulatory Flexibilities for Approval of COVID-19 Vaccines

Countries/territories employed a range of regulatory flexibilities to expedite the approval of the COVID-19 vaccines while maintaining optimal oversight of other (non-COVID-related) medical products. The current study has highlighted the flexibilities undertaken across the key domains of registration, evaluation, authorization, market surveillance, licensing, inspections, clinical trials, and lot release. These efforts were primarily stated to reduce the administrative burden, improve efficiency, avoid delays, and optimize resources, allowing timely access to quality-assured vaccines and other non-COVID-related medical products. Other studies have reported that regulatory flexibilities are critical in accelerating vaccine development and clinical trials, such as combined clinical trial phases, pre-submission discussions, and rolling submission on manufacturing validation [23,24].

Furthermore, operational and digital adaptations, such as issuing waivers for product sample submissions and remote working in selected functions, should be regarded as crucial in safeguarding the continuity of the supply chain while maintaining regulatory oversight [21,24,25,26]. Tshering et al. highlighted the essence of expedited EUA processes and interim guidelines in significantly reducing regulatory approval and licensing timelines [25]. Discussions are ongoing concerning the need for regulatory agencies to use the experiences gained during the pandemic to implement appropriate long-term regulatory efficiencies [24]. Long-term flexible regulations will also allow the agencies to adapt to the changing landscape of vaccines manufacturing and ensure timely access to essential health products during emergencies and beyond.

### 4.3. Effect of WHO EUL on the Status and Timing of COVID-19 Regulatory Approvals

As demonstrated by the rapid and substantial number of regulatory approvals and clearances, the WHO’s EULs substantially enabled a fast and coordinated response to the pandemic through an accelerated global approval process of COVID-19 vaccines. Although conducting independent assessments resulted in the longest time for approval, in some cases, fast-tracked procurements using regional assessments and other work-sharing initiatives were possible. Other scholars have also highlighted the essential role of the WHO’s EULs and other collaborative regulatory efforts in expediting global vaccine approvals [20,27,28].

Despite the observed success, a critical need remains for the continued and detailed evaluation of the approved vaccines’ safety and effectiveness to provide a more in-depth assessment/verification of the processes used to grant the EULs or EUAs [29]. Furthermore, considering the differences in long-term performance and safety profiles among the vaccine platforms, ongoing post-market surveillance methods are essential for generating real-world data. Efforts should also be made to implement more structured EUA and pre-approval access routes to address the observed inequities in vaccine access due to the existing socioeconomic and geographical barriers [30]. These findings, therefore, advocate for continued measures to modernize regulatory processes and promote international regulatory collaboration during emergencies to facilitate a rapid response and access to essential medicines.

### 4.4. Variation of COVID-19 Vaccines Approval Timelines with Wealth Status and Regulatory Maturity Levels

Higher wealth or regulatory maturity levels were associated with longer approval timelines due to complex and lengthy approval processes. These regulatory authorities are usually the first regulators to evaluate the submission, ensuring a complete evaluation of the quality, non-clinical, and clinical aspects. Hence, lengthy and stringent evaluations are needed compared to approval via reliance mechanisms. Thanks to reliance, most LMICs achieved faster approval rates because they only processed vaccine import permits [20,31]. Further, insufficient competent human resources and well-structured processes contributed to a higher reliance on external agencies and networks for the approval of vaccines among regulatory agencies in the LMICs [20,21,27,32].

However, despite the faster approvals, most LMICs were reported to have achieved lower vaccination rates due to their limited access to vaccines, geographical barriers, logistical constraints, and a lack of adequate vaccination machinery [21,30,31,33,34]. While thorough review processes are highly relevant in ensuring comprehensive quality and safety evaluations, finding a good balance between approval time, adequacy of the review, and accessibility to vaccines should be regarded as crucial in providing timely and optimal vaccination rates [21,29]. Among these measures will be to learn from the trust given to the WHO’s EULs by countries with lower wealth and regulatory maturity statuses, leading to enhanced speed in vaccine approvals during public health emergencies.

### 4.5. Adopted Best Practices and the Essence of Sustaining Them Beyond the Pandemic

Implementing a range of best practices by Member States and global partners enhanced the efficiency and effectiveness of various forms of regulatory approvals and clearances for COVID-19 vaccines. This study has revealed various best practices that resulted in better handling of multiple processes, products, personnel, and other resources while ensuring higher levels of preparedness for future public health emergencies. It is essential to note that the shorter approval timelines among most Member States were not just because of the emergency but rather the outcomes of well-structured best practices. The shorter vaccine approval times also emphasize the role of the WHO’s EULs, structured reliance mechanisms, and emergency preparedness plans in facilitating the timely approval of vaccines globally [20,21,26]. Other studies have stressed the need for platform technology, effective multi-sectoral stakeholder coordination, and strengthening regulatory systems to streamline approval processes and ensure supply chain continuity in emergencies [23,35,36,37]. Moreover, regulatory agencies should work toward institutionalizing the best practices adopted during the pandemic for more resilient and adaptive regulatory systems for higher regulatory efficiency [21,24,26].

Overall, the findings indicate a significant shift in health product regulation theory and practice, emphasizing balancing speed with scientific validity. This necessitates the creation of thorough provisions for emergency preparedness, regulatory reliance, and administrative flexibility in regulatory practices worldwide.

### 4.6. Lessons Learned: Integrating Key Insights

Responding to the COVID-19 vaccine revealed that, without proper coordination, global regulatory efforts can become disjointed and delayed. Allocations driven by donors sometimes overlooked national preparedness, highlighting the need for shared standards and respect for the local regulatory authority. To balance timely access with proper procedures, emergency frameworks must uphold confidentiality, inclusion, and transparency. Flexible resource models and partnerships established before a crisis prove essential, demonstrating that resilience depends not only on systems but also on relationships built in advance.

### 4.7. Study Limitations

This study has several limitations. Firstly, some countries did not respond, which limited how well the global representation was captured and possibly distorted the regional insights. Additionally, the analysis focused only on vaccines that received WHO Emergency Use Listing, excluding those approved solely at the national level. As a result, the study did not capture the regulatory approaches and timelines for these nationally approved vaccines, which may have underrepresented the different strategies used during the pandemic. Furthermore, some data from the regulatory authorities was self-reported, which may have been biased or inconsistent.

## 5. Conclusions

In conclusion, the global response to the COVID-19 pandemic has revealed the strengths and limitations of the existing regulatory frameworks. The WHO’s approach was instrumental in accelerating access to quality-assured COVID-19 vaccines, particularly in LMICs. Despite various challenges, the process demonstrated significant success and confirmed the feasibility of global coordination approaches. Member States must learn from the highlighted findings and strengthen regulatory frameworks to establish and sustain regulatory efficiency during public health emergencies. This includes ensuring greater independence while fostering smooth collaboration with other agencies and relevant stakeholders. On the other hand, the WHO and other global partners should develop and disseminate clear, harmonized guidelines on regulatory preparedness, such as the recently updated WHO guidelines for pandemic vaccines, which have included lessons learned from the pandemic. Moreover, practices around global health governance to secure high-level political support and clarity of international responsibilities should be strengthened.

## Figures and Tables

**Figure 1 vaccines-13-00638-f001:**
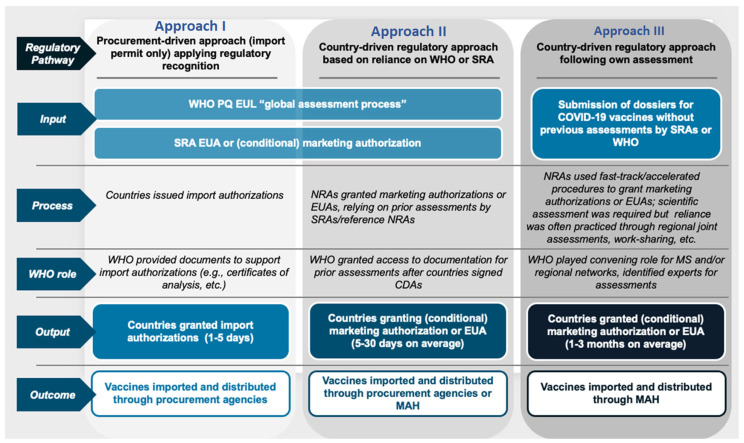
Three approaches followed by the Member States to the regulatory approval of COVID-19 vaccines during the pandemic.

**Figure 2 vaccines-13-00638-f002:**
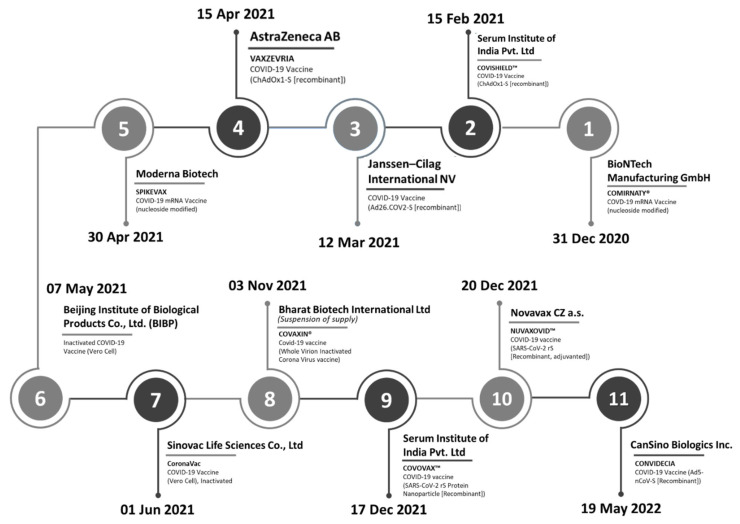
Timeline of the WHO’s Emergency Use Listings of COVID-19 vaccines.

**Figure 3 vaccines-13-00638-f003:**
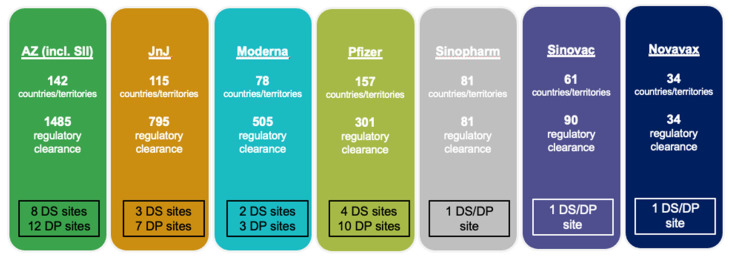
Status of member state approvals for seven COVID-19 vaccines issued EULs by the WHO (AZ = AstraZeneca, SII = Serum Institute of India, JnJ = Johnson & Johnson, DS = drug substance, DP = drug product).

**Figure 4 vaccines-13-00638-f004:**
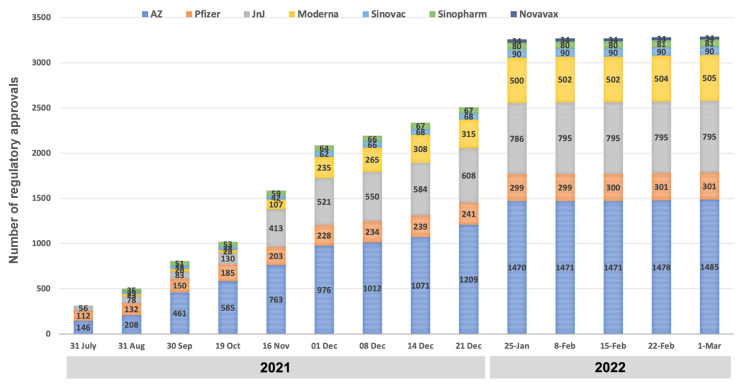
Status of regulatory approvals for seven COVID-19 vaccines in 183 Member StatesMember States, as of 1 March 2022.

**Figure 5 vaccines-13-00638-f005:**
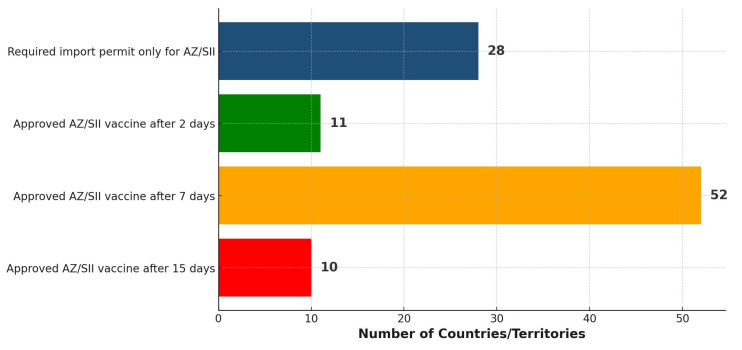
Distribution of 101 countries/territories based on the timing of approving/importing the AZ/SII vaccine within 15 days following the WHO EUL.

**Table 1 vaccines-13-00638-t001:** Details of the studied NRAs.

SN	Parameter	Classes	Frequency
1	WHO region	AFR	47
		AMR	35
		EMR	21
		EUR	54
		SEAR	11
		WPR	27
2	World Bank economic status	Low income	28
		Lower-middle income	53
		Upper-middle income	55
		High income	58
3	WHO-GBT maturity level	Level 1	111
		Level 2	25
		Level 3	33
		Level 4	25

AFR = African egion, AMR = Region of the Americas, EMR = Eastern Mediterranean Region, EUR = European Region, SEAR = South-East Asia Region, WPR = Western Pacific Region, WHO = World Health Organization, GBT = Global Benchmarking Tool.

**Table 2 vaccines-13-00638-t002:** Distribution of Member States in different WHO regions based on approaches to the regulatory approval of COVID-19 vaccines during the pandemic.

Category	AFR	AMR	EMR	EUR	SEAR	WPR
**Approach I**: Import permits only (procurement based)	11	0	1	14	1	0
**Approach II**: Approved based on reliance on expedited regulatory pathways	11	30	11	0	7	12
**Approach II**: Not yet approved but relies on at least one expedited regulatory pathway	12	0	3	1	1	4
**Approach III**: Approved based on self or collective regional evaluation	0	0	0	34	0	0

**Table 3 vaccines-13-00638-t003:** Types and key outcomes of regulatory flexibilities employed by Member States over various regulated products during the COVID-19 pandemic.

Regulatory Function	Employed Flexibility Practices	Key Outcomes
**Registration and marketing authorization (MA)**	**Submission:** Shift to e-submissions and reviews of dossiers Accept clinical trial data funded by the public sector Implement rolling submissions Accept signed copies and electronic signatures instead of original documents Streamlined requirements (“must have” versus “nice to have”) Accept “common technical dossier”	⚬ Accelerated approval timelines ⚬ Enhanced efficiency and resource optimization ⚬ Strengthened regulatory transparency and collaboration
	**Evaluation:** Expedite review and authorization Use of consortiums or work-sharing initiatives Classification of evaluation queries into critical and major Suspend time limits for response to assessment questions Implement mutual and zero-day recognition procedures as well as repeat-use protocol. Use joint evaluation	⚬ Reduced duplication and faster approvals ⚬ Optimized regulatory resources and workload sharing ⚬ Increased flexibility in regulatory approval timelines ⚬ Enhanced global harmonization and reliance
	**Authorization:** Allow justified postponement of complete dossier submission for expiring MAs facing renewal challenges. Postpone sunset clause based on justified requests Regard EUA and conditional MA as time limited and may be revoked Use of quality risk management for relaxation of post-approval changes obligations Extending the permitted implementation period	⚬ Ensures continued availability of essential medicines ⚬ Enhanced regulatory flexibility in managing marketing authorizations ⚬ Reduced administrative burden ⚬ Timely market access while ensuring compliance
**Market surveillance and control (MC)**	Waiver of pre- or post-shipment testing of imported products. Relaxation of import control procedures Use of list of medicines for exceptional importation and sale	⚬ Faster access to priority medicines ⚬ Optimized use of resources ⚬ Balanced regulatory oversight and flexibility
**Licensing of establishments (LE)**	Fast-track procedures for the certification of manufacturing facilities Extend validity of GxP compliance based on risk management principles Allow supplier/vendor requalification through risk-based approaches and reliance	⚬ Faster access to priority medicines ⚬ Balanced regulatory oversight and flexibility ⚬ Optimized use of resources ⚬ Enhanced supply chain stability ⚬ Promoted reliance and harmonization
**Regulatory inspection (RI)**	Allow distant or virtual (remote) inspections Increased reliance on inspections Allow for justified or risk-based time-limited qualification of premises and equipment Allow and rely on licensed professionals’ judgment and expertise to solely assess and make decisions on minor deviations	⚬ Flexible and streamlined regulatory processes with higher efficiency and reduced delays ⚬ Optimized use of resources ⚬ Balanced regulatory oversight and flexibility ⚬ Enhanced supply chain stability
**Clinical trials**	Expedite review and authorization of priority products for clinical trials Accept clinical trial data funded by the public sector Parallel submission of application to NRA and EC/IRB Accept multi-regional clinical trial (MRCT) data Implement reliance to CTA by other NRAs with good review system in CTA Accept an adaptive clinical trial	⚬ Faster development and access to priority medicines ⚬ Reduced reliance on industry-funded trials
**Lot release (LR)**	Allow lot release based on only review of summary lot protocol	⚬ Speeds up vaccine and biologics distribution ⚬ Reduces testing delays while maintaining safety oversight ⚬ Optimized use of resources

**Table 4 vaccines-13-00638-t004:** The Spearman’s rank correlation coefficient between the ranks reflects the country’s wealth and regulatory maturity.

Independent Variables	Dependent Variable	Spearman’s Rank Correlation Coefficient
**World Bank status ranks**	**Time of approval ranks**	*ρ* = 0.36 (two tailed *p* value = 0.00)
Low income Lower middle income Upper middle income High income	Requires import permit only for AZ/SII vaccine Approved AZ/SII vaccine after 2 days Approved AZ/SII vaccine after 7 days Approved AZ/SII vaccine after 15 days Not yet approved BUT relies on at least one expedited regulatory pathway Not approved AND does not rely on any expedited pathway	
**WHO maturity level ranks**		*ρ* = 0.51 (two tailed *p* value = 0.00)
Some elements of the regulatory system exist Evolving national regulatory system that partially performs essential regulatory functions Stable, well-functioning, and integrated regulatory system Regulatory system operating at an advanced level of performance and continuous improvement		

(Key: AZ = AstraZeneca, SII = Serum Institute of India).

**Table 5 vaccines-13-00638-t005:** Description of best practices and respective activities in approving COVID-19 vaccines.

Best Practices	Involved Activities (What to Do?)
Having an emergency preparedness plan	⚬ Facilitate pre-planning for emergencies ⚬ Ensure administrative flexibility ⚬ Prescribe the effective management of EUAs ⚬ Oversee a comprehensive product life cycle management ⚬ Enable the rapid mobilization of personnel ⚬ Allow flexibility in legal frameworks ⚬ Strong and robust regulatory system capacity
Practicing regulatory reliance on reference authorities and institutions	⚬ Avoid delays and resource strain ⚬ Facilitate efficiency in submission and evaluation processes ⚬ Improve coordination and alignment with SRAs and the WHO
Responding with flexibility	⚬ Rapid transition to digital communication ⚬ Streamline regulatory requirements ⚬ Avoid unnecessary bureaucracy and redundancy ⚬ Simplify formal approval processes for donations ⚬ Embrace flexibility in vaccine preferences ⚬ Establish a clear legal protection and liability handling path
Optimal coordination of key stakeholders	⚬ Enhance communication and decision-making efficiency ⚬ Enhance coordination with customs for efficient clearance ⚬ Strengthen community engagement and misinformation countermeasures ⚬ Combat the distribution of substandard/falsified vaccines ⚬ Address local distribution challenges ⚬ Trust and adhere to agreements under global distribution systems
Sustaining regulatory efficiency post-pandemic	⚬ Expand electronic solutions ⚬ Implement robust prioritization strategies for the optimal use of resources ⚬ Strengthen reliance on trusted authorities ⚬ Institutionalize regulatory agility ⚬ Invest in regulatory capacity building

## Data Availability

The data presented in this study are available on request from the corresponding author due to privacy/confidentiality reasons.

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
