# Peer review of "Strengthening Vaccine Regulation: Insights from COVID-19 Vaccines, Best Practices, and Lessons for Future Public Health Emergencies"

_vaccines, 2025, doi:10.3390/vaccines13060638_

Round 1
Reviewer 1 Report
Comments and Suggestions for Authors
- Interesting manuscript, about which I only have a few observations.
- In the introduction, when the acronym NRA is used, it doesn't mention what it stands for. Please mention it.
- When using open-access databases, this may be a project that does not require approval by an ethics committee; however, it would be desirable to include this information somewhere in the methods section.
- Table 1 uses many abbreviations. Please add all the abbreviations and what they refer to in a table footnote.
- Although the manuscript discusses the regulatory aspects of vaccine approval, I believe that it should be addressed in discussions that, beyond the regulatory aspects, scientific aspects are also important for vaccine approval, and that during this pandemic, in some cases, they were criticized. It should be mentioned that in some cases, approvals were rapid, but this limited the study of the long-term effects of new technologies, which should be considered in the future. I suggest adding the following references to this topic, which discuss the importance of assessing long-term effects and other aspects related to the use of RNA vectors or adenoviruses as vectors for vaccines or therapies.
- Banoun H. mRNA: Vaccine or Gene Therapy? The Safety Regulatory Issues. Int J Mol Sci. 2023 Jun 22;24(13):10514. doi: 10.3390/ijms241310514. PMID: 37445690; PMCID: PMC10342157.
- Montes-Galindo DA, Espiritu-Mojarro AC, Melnikov V, Moy-López NA, Soriano-Hernandez AD, Galvan-Salazar HR, Guzman-Muñiz J, Guzman-Esquivel J, Martinez-Fierro ML, Rodriguez-Sanchez IP, Paz-Michel B, Zaizar-Fregoso SA, Sanchez-Ramirez CA, Ramirez-Flores M, Delgado-Enciso I. Adenovirus 5 produces obesity and adverse metabolic, morphological, and functional changes in the long term in animals fed a balanced diet or a high-fat diet: a study on hamsters. Arch Virol. 2019 Mar;164(3):775-786. doi:10.1007/s00705-018-04132-6. Epub 2019 Jan 21. PMID: 30666458.
- Tsilingiris D, Vallianou NG, Karampela I, Muscogiuri G, Dalamaga M. Use of adenovirus type-5 vector vaccines in COVID-19: potential implications for metabolic health? Minerva Endocrinol (Turin). 2022 Sep;47(3):264-269. doi: 10.23736/S2724-6507.22.03797-6. Epub 2022 May 27. PMID: 35621112.
- I also believe it should be mentioned that once a vaccine is approved, surveillance should continue, as not all vaccines have the same effects. Studies showed that those vaccinated with RNA molecules had better protection than those vaccinated with the adenovirus vaccine if they unfortunately developed severe disease some time after vaccination (differences between vaccine types).
- Mendoza-Hernandez MA, Guzman-Esquivel J, Ramos-Rojas MA, Santillan-Luna VV, Sanchez-Ramirez CA, Hernandez-Fuentes GA, Diaz-Martinez J, Melnikov V, Rojas-Larios F, Martinez-Fierro ML, Tiburcio-Jimenez D, Rodriguez-Sanchez IP, Delgado-Enciso OG, Cabrera-Licona A, Delgado-Enciso I. Differences in the Evolution of Clinical, Biochemical, and Hematological Indicators in Hospitalized Patients with COVID-19 According to Their Vaccination Scheme: A Cohort Study in One of the World's Highest Hospital Mortality Populations. Vaccines (Basel). 2024 Jan 11;12(1):72. doi: 10.3390/vaccines12010072. PMID: 38250885; PMCID: PMC10821037.
- Huespe IA, Ferraris A, Lalueza A, Valdez PR, Peroni ML, Cayetti LA, Mirofsky MA, Boietti B, Gómez-Huelgas R, Casas-Rojo JM, Antón-Santos JM, Núñez-Cortés JM, Lumbreras C, Ramos-Rincón JM, Barrio NG, Pedrera-Jiménez M, Martin-Escalante MD, Ruiz FR, Onieva-García MA, Toso CR, Risk MR, Klén R, Pollán JA, Gómez-Varela D. COVID-19 vaccines reduce mortality in hospitalized patients with oxygen requirements: Differences between vaccine subtypes. A multicontinental cohort study. J Med Virol. 2023 May;95(5):e28786. doi: 10.1002/jmv.28786. PMID: 37212340.
- It is important to mention that it is necessary to continue studying the variability and long-term effects of vaccination. This is reflected, for example, in the withdrawal of the Astra vaccine for various reasons.
Author Response
We are grateful for the thoughtful and constructive feedback you provided on our manuscript (ID vaccines-3653267), “Strengthening Vaccine Regulation: Insights from COVID-19 vaccines, Best Practices, and Lessons for Future Public Health Emergencies.”
We have carefully addressed each point raised and revised the manuscript accordingly, as shown in the table below. Moreover, all changes have been highlighted in the revised manuscript.
SN |
Reviewer’s Comment |
Authors’ responses |
Page/line number |
1 |
In the introduction, when the acronym NRA is used, it doesn't mention what it stands for. Please mention it. |
Agreed, the suggested change was implemented |
Page 2, line 53-54 |
2 |
When using open-access databases, this may be a project that does not require approval by an ethics committee; however, it would be desirable to include this information somewhere in the methods section. |
Ethical review was carried out by the respective unit within the WHO |
NA |
3 |
Although the manuscript discusses the regulatory aspects of vaccine approval, I believe that it should be addressed in discussions that, beyond the regulatory aspects, scientific aspects are also important for vaccine approval, and that during this pandemic, in some cases, they were criticized. It should be mentioned that in some cases, approvals were rapid, but this limited the study of the long-term effects of new technologies, which should be considered in the future. |
Agreed, the following sentences were added under section 4.1 of the manuscript
„However, it is essential to acknowledge the importance of rigorous scientific evaluation in providing regulatory approval, a process that has faced criticism in certain contexts during the COVID-19 pandemic and deserves careful consideration in future vaccine approval assessments. It is important to note that, in some instances, the need for rapid authorization has restricted the ability to evaluate the long-term effects of new vaccine platforms, highlighting a crucial area for consideration in future emergency-use guidelines and post-market surveillance plans.“ |
Page 14, Lines 298 - 304 |
4 |
It should also be mentioned that once a vaccine is approved, surveillance should continue, as not all vaccines have the same effects. Studies showed that those vaccinated with RNA molecules had better protection than those vaccinated with the adenovirus vaccine if they unfortunately developed severe disease some time after vaccination (differences between vaccine types). It is important to mention that it is necessary to continue studying the variability and long-term effects of vaccination. This is reflected, for example, in the withdrawal of the Astra vaccine for various reasons |
Agreed, the following sentence was added under section 4.3:
“Furthermore, considering the differences in long-term performance and safety profiles among vaccine platforms, ongoing post-market surveillance methods are essential for generating real-world data“ |
Page 15, Lines 338 - 341 |
Reviewer 2 Report
Comments and Suggestions for Authors
This manuscript delineates a comprehensive assessment of regulatory strategies adopted by 194 National Regulatory Authorities (NRAs) for COVID-19 vaccine approval between February 2021 and March 2022. The study is well-structured and provides valuable insights into the global regulatory landscape during the pandemic, and lessons learned to inform preparedness for future public health emergencies.
Comments.
1. Funding transparency:
Was this study supported by a specific WHO funding or grant number?
Consider including the funding reference number for transparency if a funding source exists.
If not, no changes are needed.
2. Inconsistencies in study period reporting:
There is an apparent inconsistency regarding the time period covered by the study
- In Section 2.2 Study Setting and Participants, the study is said to cover February 2021 to March 2022.
- In Section 3.3 Status and Timing of Member State Approvals, the WHO EUL timeline spans December 2020 to May 2022.
Additionally, several countries had already emergency use authorisation of COVID-19 vaccines prior to February 2021, such as:
- Russian Federation: Gam-COVID-Vac on 11 August 2020 ( https://sputnikvaccine.com/newsroom/pressreleases/sputnik-v-vaccine-granted-full-permanent-approval-in-russia/ )
- United Kingdom: BNT162b2 on 2 December 2020 ( https://www.gov.uk/government/news/uk-medicines-regulator-gives-approval-for-first-uk-covid-19-vaccine )
- Bahrain: BNT162b2 on 4 December 2020 ( https://www.aljazeera.com/news/2020/12/4/bahrain-becomes-second-country-to-approve-pfizer-covid-19-vaccine )
- United States: BNT162b2 on 11 December 2020 ( https://www.fda.gov/news-events/press-announcements/fda-takes-key-action-fight-against-covid-19-issuing-emergency-use-authorization-first-covid-19 )
Suggest clarifying why the study’s analytical window begins in February 2021. Was this the start of systematic data collection?
If earlier data were excluded, explain how this may have affected comprehensiveness.
3. Data reliability and completeness:
While the study includes data from 194 NRAs, but it is unclear whether all countries provided complete and high-quality data/sources.
Suggest clarifying the data completeness by WHO region or income group.
Were there any gaps in reporting, especially from conflict-affected areas (e.g., Gaza & West Bank, Kashmir) or remote island states/territories?
If relevant, acknowledge this as a limitation and discuss how it might affect generalisability.
4. Impact of Political and Economic factors beyond the NRA maturity:
The manuscript briefly mentions that factors beyond NRA maturity may influence approval timelines (lines 282–284), but these potential factors do not elaborate.
I suggest expanding on external factors that may have affected vaccine regulation pathways, such as Vaccine nationalism, Geopolitical preferences or alignments, and Bilateral procurement outside the COVAX.
Errors.
1. Line 44:
Current: “such as personal protective equipment (PPE) and diagnostics, vaccines...”
Suggested revision: “such as personal protective equipment (PPE), diagnostics, and vaccines...”
Author Response
We are grateful for the thoughtful and constructive feedback you provided on our manuscript (ID vaccines-3653267), “Strengthening Vaccine Regulation: Insights from COVID-19 vaccines, Best Practices, and Lessons for Future Public Health Emergencies.”
We have carefully addressed each point raised and revised the manuscript accordingly, as shown in the table below. Moreover, all changes have been highlighted in the revised manuscript.
SN |
Reviewer’s Comment |
Authors’ responses |
Page/line number |
1 |
This manuscript delineates a comprehensive assessment of regulatory strategies adopted by 194 National Regulatory Authorities (NRAs) for COVID-19 vaccine approval between February 2021 and March 2022. The study is well-structured and provides valuable insights into the global regulatory landscape during the pandemic, and lessons learned to inform preparedness for future public health emergencies. |
Authors would like to thank the reviewers for the very positive feedback and highlights on the value brought about by our study into the global regulatory landscape during public health emergencies. |
NA |
2 |
Funding transparency: Was this study supported by a specific WHO funding or grant number? Consider including the funding reference number for transparency if a funding source exists. If not, no changes are needed. |
The study was not supported by a specific WHO funding or grant number but rather a mixture of contributing partners channelling the financial support to the WHO for different causes including evidence generation. |
NA |
3 |
Inconsistencies in study period reporting: There is an apparent inconsistency regarding the time period covered by the study
- In Section 2.2-Study Setting and Participants, the study is said to cover February 2021 to March 2022. - In Section 3.3 Status and Timing of Member State Approvals, the WHO EUL timeline spans December 2020 to May 2022.
|
We thank the reviewer for this observation. The typographical error on section 3.3 has been corrected to read March 2022. Notably, the study was initiate (Feb 2021) after the WHO EULs had already started (Dec 2020). |
Page 8, line 192
|
4 |
Additionally, several countries had already emergency use authorisation of COVID-19 vaccines prior to February 2021, such as:
· Russian Federation · United Kingdom · Bahrain · United States
I Suggest clarifying why the study’s analytical window begins in February 2021. Was this the start of systematic data collection? If earlier data were excluded, explain how this may have affected comprehensiveness.
|
We agree with the reviewer’s observation that some countries had already Emergency Use Authorisation (EUA) of COVID-19 vaccines before the inception of this study.
However, the study focused on only the vaccines which were approved by Member states after being approved through the WHO’s EULs as part of the COVAX initiative and not those which were approved via country specific initiatives. This has been captured as among the limitation of this study
|
Page 16, lines 396 - 404 |
4 |
Data reliability and completeness: While the study includes data from 194 NRAs, but it is unclear whether all countries provided complete and high-quality data/sources.
Suggest clarifying the data completeness by WHO region or income group. Were there any gaps in reporting, especially from conflict-affected areas (e.g., Gaza & West Bank, Kashmir) or remote island states/territories?
If relevant, acknowledge this as a limitation and discuss how it might affect generalisability.
|
We appreciate this valuable comment from the reviewer. We aimed and reached out to 194 NRAs to participate in this study. However, data on approaches undertaken for regulatory approvals of COVID-19 vaccines were obtained from a total of 153 NRAs/countries as indicated in their respective WHO-regions on Table 2.
Moreover, data regarding the status of approvals of the 7 COVID-19 vaccines was successfully obtained from a total of 183 member states as described on Figure 4 of the manuscript.
This has now been highlighted as among the limitations of our study on section 4.7. Based on the large number and good distribution of the member states which responded, findings from this study are highly generalizable.
|
Page 16, lines 396 - 404 |
5 |
Impact of Political and Economic factors beyond the NRA maturity: The manuscript briefly mentions that factors beyond NRA maturity may influence approval timelines (lines 282–284), but these potential factors do not elaborate. I suggest expanding on external factors that may have affected vaccine regulation pathways, such as Vaccine nationalism, Geopolitical preferences or alignments, and Bilateral procurement outside the COVAX.
|
Agreed, the suggested additional factors have been included |
Page 14, lines 285 - 286 |
6 |
Errors. 1. Line 44: Current: “such as personal protective equipment (PPE) and diagnostics, vaccines...” Suggested revision: “such as personal protective equipment (PPE), diagnostics, and vaccines...”
|
Agreed, the suggested revision has been implemented |
Page 2, line 44 |
Reviewer 3 Report
Comments and Suggestions for Authors
This is a very interesting and scientifically rigorous study. The paper is very well organized and written. I only noted that the use of the acronym ‘National Regulatory Authorities (NRAs)’ that appears in the abstract should also be used in the introduction when it first appears (line 66).
Graphs and figures are of great importance to make the findings visible and facilitate the reading of the study.
I also submit to the authors' consideration whether the section on seven lessons learned should be included in the discussion section, highlighting its content, given the great relevance of these findings
A section on strengths and limitations should be included. Are there any limitations to collecting information through a self-administered questionnaires?
Author Response
We are grateful for the thoughtful and constructive feedback you provided on our manuscript (ID vaccines-3653267), “Strengthening Vaccine Regulation: Insights from COVID-19 vaccines, Best Practices, and Lessons for Future Public Health Emergencies.”
We have carefully addressed each point raised and revised the manuscript accordingly, as shown in the table below. Moreover, all changes have been highlighted in the revised manuscript.
|
Reviewer’s Comment |
Authors’ responses |
Changes on page/line number |
1 |
This is a very interesting and scientifically rigorous study. The paper is very well organized and written. I only noted that the use of the acronym ‘National Regulatory Authorities (NRAs)’ that appears in the abstract should also be used in the introduction when it first appears (line 66). |
We thank the reviewer for the positive feedback on our study.
The phrase National Regulatory Authorities was added on the introduction where it first appeared |
Page 2, Line 53 |
2 |
I also submit to the authors' consideration whether the section on seven lessons learned should be included in the discussion section, highlighting its content, given the great relevance of these findings. |
Agreed, the section discussing the aspects from the seven lessons learned has been included in the discussion section. |
Page 16, lines 387 - 395 |
3 |
A section on strengths and limitations should be included. Are there any limitations to collecting information through self-administered questionnaires? |
Agreed, a section on limitations of the study has been included to capture encountered limitations such as non-responses from some countries, considering only vaccines which received EULs from the WHO hence lacking the data on vaccines that were only nationally approved |
Page 16, lines 396 - 404 |